# Continuum and Molecular Modeling of Chemical Vapor Deposition at Nano-Scale Fibrous Substrates

**Himel Barua †** **and Alex Povitsky ***

Department of Mechanical Engineering, College of Engineering, University of Akron, Akron, OH 44325, USA; baruah@ornl.gov
* Correspondence: povitsky@uakron.edu
† Present affiliation: The U.S. Department of Energy (DOE), Oak Ridge National Laboratory (ORNL), 1 Bethel Valley Rd 5200, Oak Ridge, TN 37830, USA.

**Abstract:** Chemical vapor deposition (CVD) is a common industrial process that incorporates a complex combination of fluid flow, chemical reactions, and surface deposition. Understanding CVD processes requires rigorous and costly experimentation involving multiple spatial scales, from meters to nanometers. The numerical modeling of deposition over macro-scale substrates has been conducted in the literature and results show compliance with experimental data. For smaller-scale substrates, where the corresponding Knudsen number is larger than zero, continuum modeling does not provide accurate results, which calls for the implementation of molecular-level modeling techniques. In the current study, the finite-volume method (FVM) and Direct Simulation Monte Carlo (DSMC) method were combined to model the reactor-scale flow with CVD around micro- and nano-scale fibers. CVD at fibers with round cross-sections was modeled in the reactor, where fibers were oriented perpendicularly with respect to the feedstock gas flow. The DSMC method was applied to modeling flow around the matrix of nano-scale circular individual fibers. Results show that for smaller diameters of individual fibers with the same filling ratio, the residence time of gas particles inside the fibrous media reduces, and, consequently, the amount of material surface deposition decreases. The sticking coefficient on the fibers' surface plays an important role; for instance, increasing the sticking coefficient from 20% to 80% will double the deposition rate.

**Keywords:** chemical vapor deposition; low-pressure reactor; direct simulation Monte Carlo; computational fluid dynamics; carbon deposition

## 1. Introduction

Chemical vapor deposition (CVD) is a material deposition process with a vapor-to-solid chemical reaction at a rigid surface. The CVD is generally used in the industrial processes of manufacturing multi-layer materials to create a thin film over a substrate surface [1]. Fluid flow through and around bundles of fibers needs to be modeled to obtain the deposition rate at the fibers' surface. The set-ups of deposition substrate surfaces vary as the surfaces could be smooth or rough, and represent isolated fibers or the collective surface of bundles of fibers [2]. Depending on the shape and size of the substrate, the CVD process involves complex no-slip or slip fluid flow phenomena along with material deposition, which makes the fluid–solid system quite complicated to model and to understand. The scope of the study is to model the CVD of carbon at the fibers' surface (Figure 1) that is qualitatively similar to filtration and physical vapor deposition processes in which solid particles deposit at the fiber surface.

Though for macro-length scales, the traditional Newtonian and continuum description of fluid mechanics with no-slip boundary conditions can resolve the gas-flow phenomena coupled with CVD process, for a smaller-length scale the major discrepancy [3] occurs between continuum approximation and real molecular flows, which causes substantial

variation in the CVD rate. To understand the validity of continuum approximation, a well-established non-dimensional group known as the Knudsen number (*Kn*) is used. The *Kn* represents the degree of rarefaction of gas by the ratio of the gas molecular mean free path over the geometric length scale of a substrate:

$$Kn = \frac{\lambda}{L},$$ (1)

where $\lambda$ is the gas molecular mean free path and $L$ is the length scale of the system, such as the diameter of an individual fiber or of a bundle of fibers.

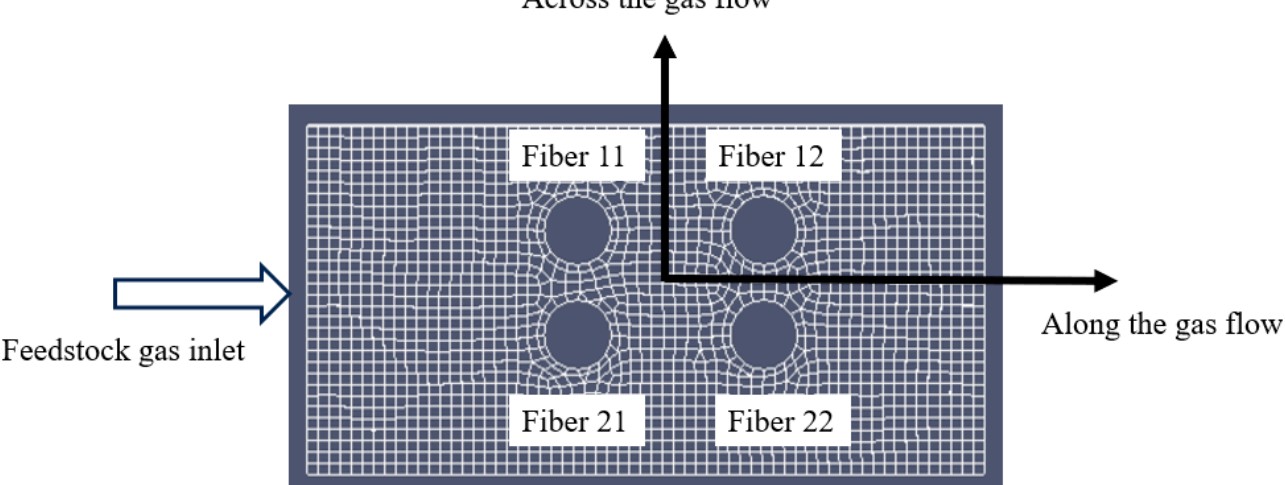

**Figure 1.** Schematic of DSMC modeling for regular fibers with DSMC cell set-up. For fibers' subscripts, the first number denotes the row of fibers, and the second number denotes the column of fibers. For instance, "Fiber21" means fiber belonging to the 2nd row and 1st column in the bundle.

In general, gas flows encounter compressibility and viscous heating. In the current case, *Ma* << 0.3, and therefore, compressibility and aerodynamic (viscous) heating are not substantial. The Mach number is the ratio of reactor flow velocity and the speed of sound. The speed of sound is evaluated at the feedstock gas temperature of 816 K in a heated reactor [4].

For a macro-scale continuum flow (*Kn* < 0.01), solving Navier–Stokes equations with no-slip boundary conditions produces solutions coinciding fairly with experiments. For smaller linear scales typical for crossflow-oriented long fibers of micron-scale diameters (0.01 < *Kn* < 0.1), the above approach becomes less accurate, and for larger Knudsen numbers (*Kn* > 0.1) the no slip approach is inadequate.

In the vapor deposition process, the gas rarefaction effect creates a thin flow layer near the rigid surface, named the Knudsen layer, which is of the linear scale of a single molecular mean free path. The Knudsen layer cannot be analyzed by using Navier–Stokes equations and requires solution of the Boltzmann equation for the local velocity distribution of molecules. For *Kn* < 0.1, the Knudsen layer covers less than 10% of the channel height for internal flows. For *Kn* < 0.1, the Knudsen layer can be neglected by extrapolating continuous bulk gas flow up to the rigid surface, which results in a finite slip velocity at the wall. This flow regime is called the partial slip flow regime.

For larger *Kn* numbers, the above continuum approach is no longer accurate and a solution of flow field in the Knudsen layer (Figure 2) is required, which can be achieved using molecular Direct Simulation Monte Carlo (DSMC). In the current research, a methodology was developed to approach problems with a large range of linear flow scales, from the reactor scale to individual fibers of a sub-micron diameter. Two well-established methods, namely, the continuum Finite Volume Method (FVM) for larger linear scales and the

molecular Direct Simulation Monte Carlo (DSMC) for flow near fibers, were combined and one-way coupled.

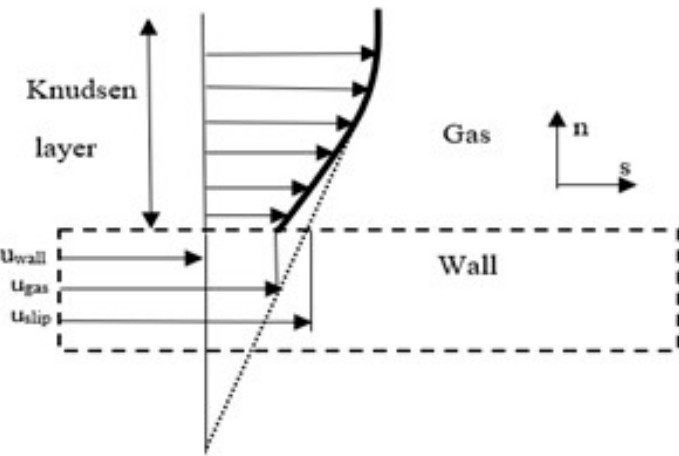

**Figure 2.** Knudsen layer and slip velocity at wall, where *n* and *s* are normal and tangential directions with respect to solid wall.

To approach the problem of determining the slip velocity in the current study, the fluid motion was considered at a molecular scale at least for a part of the domain instead of considering the fluid as a continuous medium. On a molecular scale, the methods that have been used by researchers are categorized as molecular dynamics (MD) and DSMC. When using MD, every computational particle represents an individual fluid molecule. This approach is more applicable for liquids at the nanoscale, where molecules always interact with their neighbors through potential inter-molecular forces. The DSMC allows a fraction of needed computing time compared to MD, as DSMC particles represent billions of physical gas molecules which interact with each other only during their collision. Statistical sampling is used to obtain fluid dynamics properties from DSMC results.

A considerable amount of research has been performed to model CVD process. While CVD modeling over a flat plate has been considered in numerous papers, for example, refs. [5,6], CVD over more complex and multi-scale geometries has drawn much less attention and has been limited to near-atmospheric-pressure reactors [7]. Most of the published approaches have modeled the kinetics of chemical reactions, while the link between the CVD modeling of coating over a single fiber surface has been investigated [8], in which an adaptive finite element method was used to capture the thickness of the deposited material. Ref. [9] conducted an experiment and continuous CFD modeling to investigate the growth process of a multi-walled carbon nanotube by CVD. In Ref [9], the authors used FVM implemented in FLUENT and studied the flow properties. Most of the modeling of carbon deposition on fibers [10–12] is limited to either flow modeling around a single fiber [13] or a reactor bulk flow modeling [14,15]. In the current research, the effects of multiple fibers, which are forming arrays of fibers, have been investigated to discover their critical role in spatial and temporal deposition profiles and the final thickness of the deposited layer of material.

DSMC has been implemented in prior studies [16,17]. Ref. [18] combined FVM with DSMC to model a chemically neutral flow around a fibrous medium. Refs. [19,20] extended DSMC to model adsorption and desorption in porous media. Ref. [21] adopted total collision energy and a quantum kinetic model [22] to implement chemical reactions in a DSMC method. The current research implemented an extension of the DSMC method to solve carbon deposition over wall surfaces. Instead of the direct modeling of volumetric and surface reactions, the proposed approach was implemented by taking the combination of absorbing and reflecting boundary conditions at the substrate wall for DSMC particles embedded in fluid flow. The sticking coefficient was adopted to account for the probability of the DSMC particle undergoing deposition. In general, the magnitude of the sticking

coefficient is a function of the temperature, surface coverage, and structural details of the surface. This approach reduces the complexity of DSMC algorithm by simplifying the surface chemical kinetics model and still obtains the local rate of adsorption of particles over the wall surface.

In the CVD process, the surface reaction is the reason particles are deposited over the surface, in which the time available for the reaction is important. The residence time indicates how long it would take for simulated particles to move from the inlet to the outlet of the DSMC domain. One of the major objectives of the current study is to characterize the residence time of particles inside the fibrous medium for different fibers' set-ups and for the range of sticking coefficients at the fiber surface.

In a broader sense, several important and widespread technologies, including Micro-electromechanical systems (MEMS), have relevant broad fluid flow scales ranging from meters to nanometers in the frame of a single set-up [23]. The common major challenge in the modeling of fluid flow in these processes is the range of fluid flow scales that requires the combined use of continuum and molecular approaches [24].

The study is composed as follows. In Section 2, the DSMC code is developed and validated using a well-known benchmark named Couette flow. The degree of gas rarefaction is represented by the Knudsen number influencing the flow velocity profile. In Section 2.1, the effect of rarefaction of CVD gas is briefly presented. The velocity profile for a range of Knudsen numbers is validated in Section 2.2 by comparing it to prior Lattice Boltzmann Method (LBM) results [25]. In Section 3, the FVM results of flowfield computations for the entire CVD reactor are presented, and are used as the inlet velocity for the DSMC domain. In Sections 3.2–3.5, the developed in-house DSMC code is used to model the particle deposition rate and residence time of particles around the fibers' substrates (Figure 1). For the DSMC simulations presented in Sections 3.3 and 3.4, infinitely long fibers with circular cross-sections in a regular orientation perpendicular to gas flow are computed for the range of Knudsen numbers, with varying distances between two consecutive fibers, by (i) changing the volume fraction occupied by fibers and (ii) keeping the volume fraction constant but varying the fibers' diameter. For both cases, the residence time of flow particles were calculated, and represents the amount of time the CVD feedstock gas molecules stay inside the fibrous media. In Section 3.5, the effect of rarefaction on the CVD is studied.

## 2. DSMC and Deposition at Rigid Surface

### 2.1. Development of DSMC and Continuum Models for Gas Flow in Proximity of Rigid Boundary

In the frame of the partial slip flow approach, $0.01 < Kn < 0.1$, the continuum approximation is still valid apart from the Knudsen layer [26]; see Figure 2. To account for the rarefaction feature at the wall, special boundary conditions are considered that are associated with the velocity slip, named Maxwell's boundary condition [27,28].

The first-order slip velocity boundary condition for a stationary wall developed by Maxwell [24,26] was used in the current study:

$$u_s = \lambda \frac{2 - \sigma_u}{\sigma_u} \frac{\partial u}{\partial n}, \tag{2}$$

where $u_s$ is the slip velocity, u is the gas velocity in the Knudsen layer next to the wall, $\lambda$ is the mean free path, and $\frac{\partial u}{\partial n}$ represents the slope of the velocity normal to the wall. The $\sigma_u$ term in Equation (2) represents the tangential accommodation coefficient. Momentum and energy transfer between gas molecules and the surface require the specification of interactions between impinging gas molecules and the surface. From a microscopic point of view, this approach is quite complicated and requires detailed knowledge of scattering kernels [24], while from a macroscopic point of view, the accommodation coefficient, $\sigma_u$, is accurate enough to describe the gas–wall interactions. For the limit case, $\sigma_u = 0$, gas molecules reflect from the surface specularly and pre-collision and post-collision velocities of gas particles are symmetric with respect to the normal direction to the surface, n. For

larger values of the accommodation coefficient $\sigma_u$, reflections of gas molecules over the wall become more random, which represents diffusive reflection.

For flows with $Kn > 0.1$, constitutive laws that define stress tensor and heat flux vectors break down [29], which requires higher-order corrections to the constitutive laws, resulting in Burnett or Woods equations [24] that are challenging to solve. In the current research, the DSMC methodology was implemented to compute slip velocity and to calculate the residence time of CVD gas molecules near a representative set of fibers. The DSMC approach comprises five major steps [3,17]:

1. Particle initialization;
2. Computational displacement of simulated particles after time interval $\Delta t$;
3. Indexing and cross referencing of simulated particles as they move between DSMC cells;
4. Computing of particles' new velocities after inter-particle collisions and collisions between simulated particles and wall surface;
5. Sampling simulated DSMC particles to obtain macroscopic properties.

The first step of DSMC modeling is initialization of the DSMC particles, which defines the properties of the fluid, the DSMC cell size, the number of DSMC cells and the number of real gas molecules represented by DSMC particles. Simulated particles in DSMC are initialized with random velocity and a random initial position. To evaluate the number of simulated particles using DSMC, let us consider a domain of volume Vc, in which molecular number density is n. The total number of real molecules in the domain is equal to nVc.

The total number of simulated particles $N = \frac{nV_c}{F_n}$, where $F_n$ is the number of real molecules represented by each simulated particle. In the current study, each simulated particle represented $10^{12}$ gas molecules. The assumption behind DSMC is that molecules have Maxwell–Boltzmann distribution everywhere in the domain and the collision between molecules are elastic in the DSMC domain. The assigned initial position and initial velocity of each DSMC particle were generated by a random number generator.

For each DSMC particle, equations of motion were solved to obtain a new location of particles at the end of the current, *n*-th, time step.

$$\frac{\mathrm{d}r_i}{\mathrm{d}t} = V_i, i = 1, \dots, N, \tag{3}$$

where $V_i$ is the speed of the *i*-th DSMC particle

During the movement stage of DSMC particles, no collisions of particles are assumed. The position of particles is updated by Equation (4):

$$r_i^{n+1} = r_i^n + \Delta t * V_i^n \tag{4}$$

After moving DSMC particles over a time step $\Delta t$, particles are sorted and indexed. Indexing serves the purpose of figuring out which particles are located in which cell. Indexing is necessary because particles might move to the next cell. The new cell locations of the molecules are indexed so that the intermolecular collisions and flow field sampling can be held accurately.

Indexing is also necessary to pick up the collision partner. In each cell, colliding particles are randomly selected. For each pair, pre-collisional velocities are replaced by post-collisional velocities for use in Equations (3) and (4) in the next time step. The proper choice of collision partner and choosing the right number of collisions are crucial to the consistency of DSMC. In the current study, the no-time-counter (NTC) method [30] was used to model the collision. This method uses a multistep collision model. At each time moment, n, only particles which reside in the same DSMC cell participate in collisions.

Every collision is considered a random event with some probability of collision in every cell. The probability of collision between particles is chosen as follows:

$$P_{coll}[i,j] = \left(\frac{1}{2V_c}\right)(F_N N)(N-1)(\sigma_T C_r)_{max}\Delta t,\tag{5}$$

where $V_c$ is the cell volume, $(\sigma_T C_r)_{max}$ is the maximum product of the collision cross section and the relative speed of all possible particle pairs in the cell, $\Delta t$ is the time step and $F_N$ is the number of real molecules or atoms that each DSMC particle represents. Particle $i$ is chosen randomly from all particles in the cell and $j$ is chosen from the same cell to ensure near-neighbor collision. The collision pair would be selected if the condition below is satisfied:

$$\frac{(\sigma_T C_r)_{ij}}{(\sigma_T C_r)_{max}} > R_f,\tag{6}$$

where $R_f$ is a random number uniformly chosen at interval [0, 1].

In the third stage of DSMC methodology, boundary conditions are implemented. In the current study, the model had inlet, outlet, and symmetrical boundaries (see Figure 1), which is also referred to as a permeable boundary condition. For the inlet, the most probable average speed is taken from the FVM solution for flow velocity inside the reactor, as outlined in Section 3.1. The inlet boundary is located 4 mm upstream from the DSMC domain (see Section 3 and Figure 7). When DSMC particles exit through the upper and lower boundaries of the DSMC domain (see Figure 1), a new DSMC particle is introduced at the opposite boundary at the exit point.

To model the deposition at the substrate surface, a novel approach was implemented by modifying the Maxwell boundary condition of diffuse scattering. A reactive coefficient or sticking coefficient [31] is introduced, which represents the probability of molecules colliding with the substrate to undergo a surface reaction process, which in CVD modeling is deposition. When a simulated particle reaches the substrate surface, a number $\eta$ is generated by the random number generator. This number is compared with the sticking coefficient. If the value of $\eta$ is lower than the sticking coefficient, the particle would deposit on the wall. If the value is higher than the sticking coefficient, the particle would reflect onto the fluid domain following the cosine rule. The higher the value of the sticking coefficient, the larger the number of deposited particles would be.

If the DSMC particle is reflected by the surface, the particle post-reflection velocity ($v_x$, $v_y$, $v_z$) is assigned as follows:

$$v_x = \sqrt{-2\left(\frac{k}{m}\right)T_w\log(\alpha_1)}\cos(2\pi\alpha_2),$$

$$v_y = \sqrt{-2\left(\frac{k}{m}\right)T_w\log(\alpha_2)},\tag{7}$$

$$v_z = \sqrt{-2\left(\frac{k}{m}\right)T_w\alpha}\sin(2\pi\alpha_2)$$

where $\alpha_1$ or $\alpha_2$ are the random numbers taken from 0 to 1. In Section 3.3, the effect of the magnitude of the sticking coefficient $\eta$ on the deposition rate will be explored.

## 2.2. Validation of DSMC Method and Computer Code

The developed DSMC code was used to model the Couette flow for validation purposes (see Figure 3). The Couette flow is the viscous flow between parallel infinite plates, where the upper plate is moving with uniform speed and the bottom plate is fixed. Couette flow is a shear flow driven by the upper plate movement in the flow direction. DSMC results were validated against the Lattice Boltzmann method (LBM) results developed by Refs. [25,32,33] for Couette flow with no-slip and partial-slip boundary conditions.

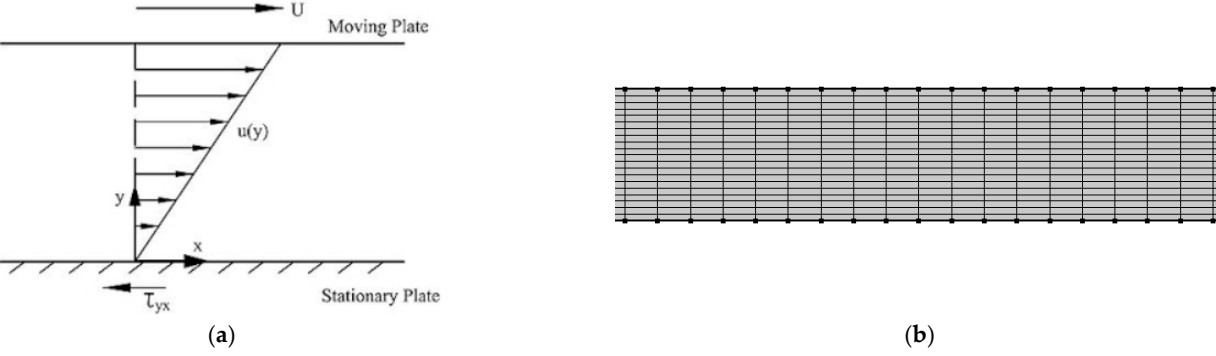

**Figure 3.** Validation of DSMC code: (**a**) Couette flow with no-slip boundary conditions and (**b**) DSMC domain with cells.

For validation purposes, the distance between two plates is 1 μm where the fluid domain is filled with the Argon gas [21,34]. At a standard temperature and pressure, the average molecular diameter is $3.66 \times 10^{-10}$ m, and the molecular mean free path is 65 nm. The plate is moving at the speed of 150 m/s (Ma~0.5) along the positive *x* direction.

To model the Couette flow, a DSMC simulation domain was created (see Figure 3b). The number of cells along the flow direction (horizontal) was 20 and the number of cells along the height was 20. The length-to-height ratio of each cell was 5. The simulation time step was chosen in such a way that each DSMC particle would stay for at least three time steps in a single cell during the simulation. Here, the time step was chosen as $2.96 \times 10^{-11}$ s and the number of time steps was 1000. If $Kn = 0.001$, the flow could be assumed to be a continuum. The pressure was $4.89 \times 10^{-6}$ Pa and the number density of gas was $1.19 \times 10^{27}$ (see Table 1). The value of $Kn = 0.01$ is a transitional Knudsen number, beyond which the gas flow starts to deviate from continuum approximation. In Table 1, the number density of gas and the corresponding pressure are listed for all the Knudsen numbers.

**Table 1.** DSMC parameters for Couette flow for various Knudsen numbers.

| Knudsen Number, *Kn* | Length (m) | Mean Free Path (m) | Pressure, Pa | Number Density, *n* |
|---|---|---|---|---|
| 0.001 | $1 \times 10^{-6}$ | $1 \times 10^{-9}$ | $4.89 \times 10^{6}$ | $1.19 \times 10^{27}$ |
| 0.01 | $1 \times 10^{-6}$ | $1 \times 10^{-8}$ | $4.89 \times 10^{5}$ | $1.19 \times 10^{26}$ |
| 0.1 | $1 \times 10^{-6}$ | $1 \times 10^{-7}$ | $4.89 \times 10^{4}$ | $1.19 \times 10^{25}$ |
| 1 | $1 \times 10^{-6}$ | $1 \times 10^{-6}$ | $4.89 \times 10^{3}$ | $1.19 \times 10^{24}$ |
| 10 | $1 \times 10^{-6}$ | $1 \times 10^{-5}$ | $4.89 \times 10^{2}$ | $1.19 \times 10^{23}$ |

To study the effect of the Knudsen number on the velocity profile, the range of Knudsen numbers was considered and for each case the velocity profile, $u(y)$, was computed by the DSMC code. The Knudsen number in the domain can be controlled by the mean free path, pressure, and number density of gas. The velocity profile and slip velocity are shown in Figures 4 and 5, while the comparison to prior LBM results is shown in Figure 6.

As shown in Figure 4a,b, the Couette flow velocity profile was obtained for the range of Knudsen numbers corresponding to a different degree of rarefaction. The flow with $Kn > 0.01$ represents partial slip flow, in which the velocity at the lower wall is small but not equal to zero. When increasing of the value of the Knudsen number, flow became more rarefied and near the bottom rigid surface the fluid velocity started to increase. In Figure 4a, for $Kn = 0.001$, the flow velocity at the lower wall was 0.05 times the upper-plate velocity, while for $Kn = 0.1$, the ratio increased to 0.12. In Figure 4b, for $Kn = 1$, the wall fluid velocity was 0.28 times the upper plate velocity and for $Kn = 10$, where the flow was quite rarefied, the wall fluid velocity was 0.32 times the upper plate velocity.

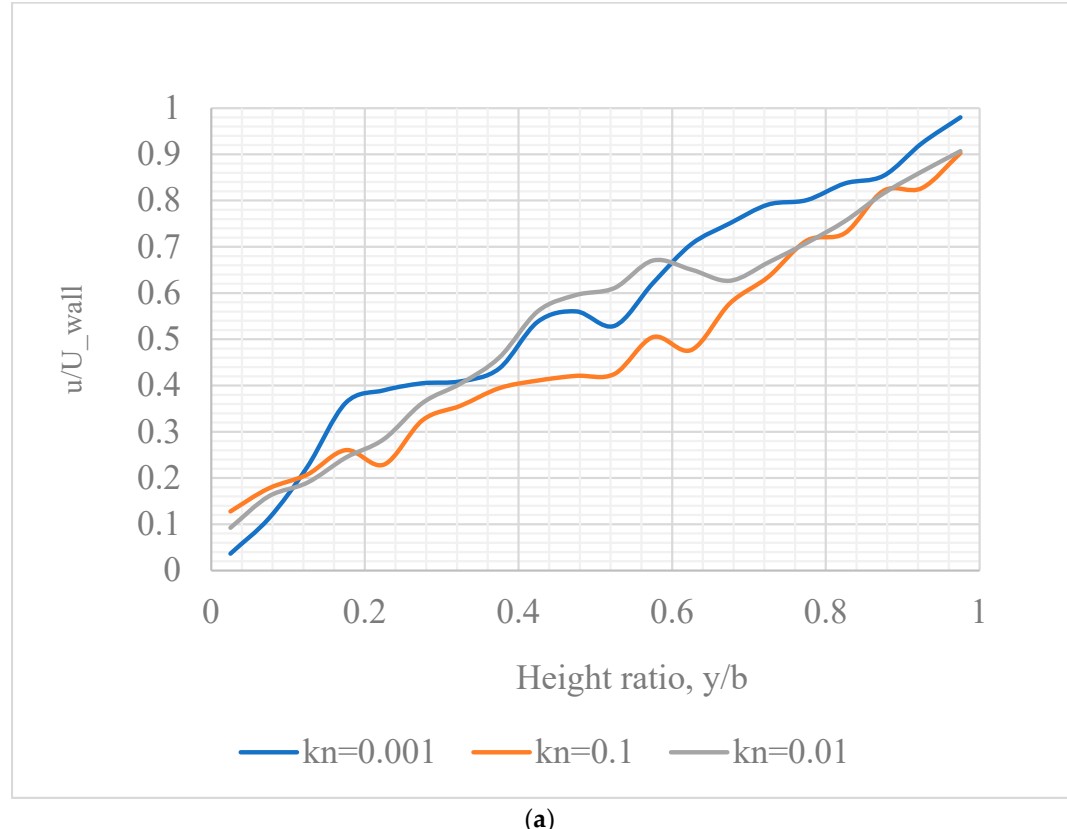

(**a**)

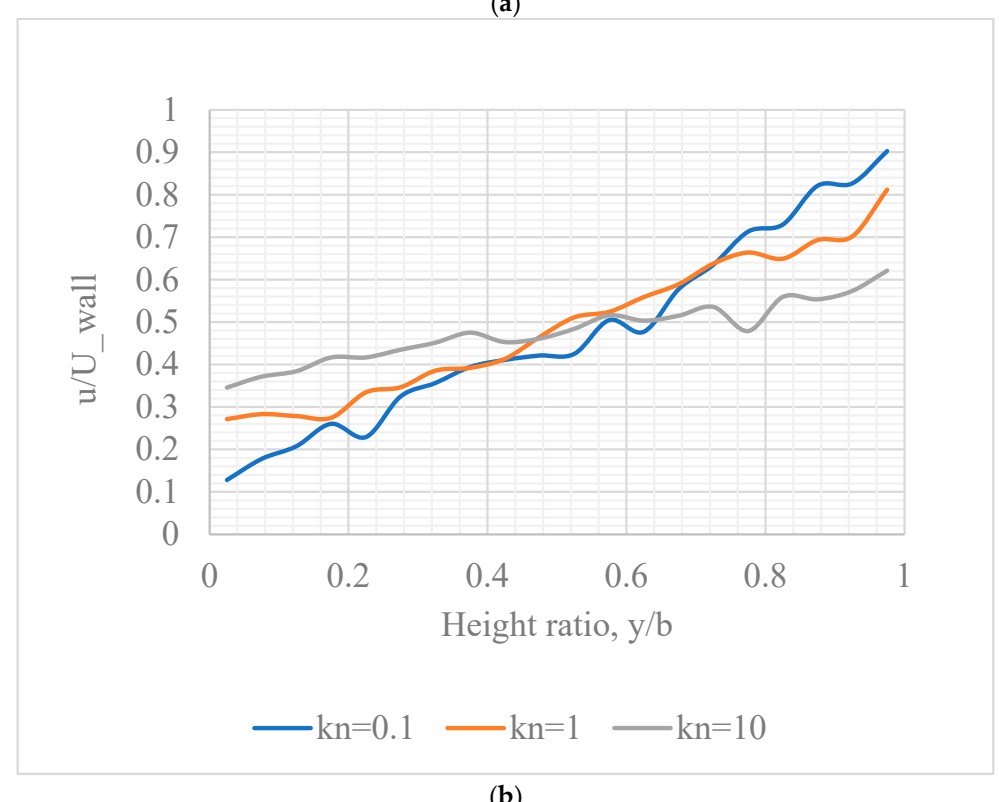

(**b**)

**Figure 4.** Couette flow: velocity profile for Knudsen number ranging (**a**) from 0.01 to 0.1 and (**b**) from 0.1 to 10. Velocity profile, $u/U_{\text{wall}}$, is presented as a function of coordinate $y$ normalized by the distance b between plates. The number of simulated particles is equal to 20 per DSMC cell.

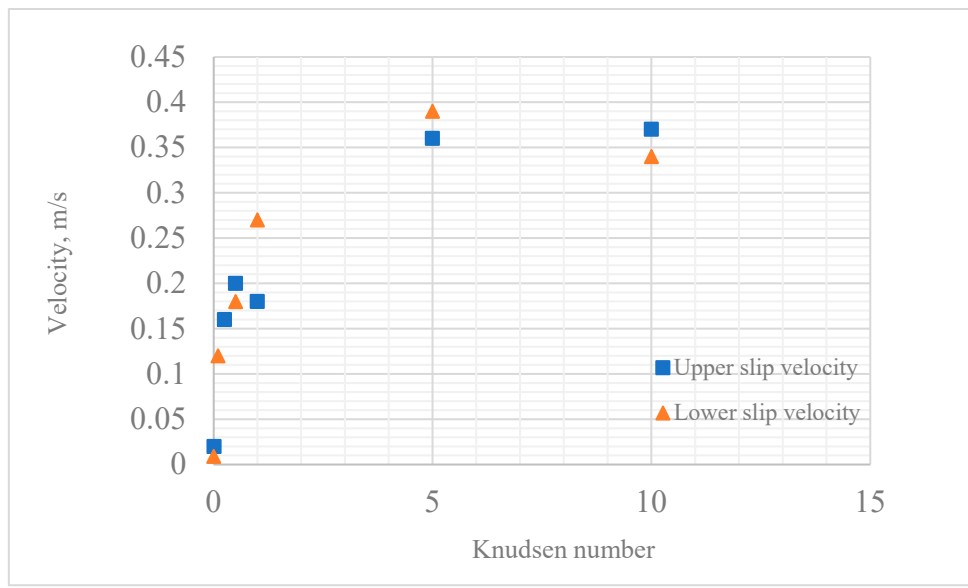

**Figure 5.** Slip velocity for Couette flow at lower and upper plates as a function of Knudsen number.

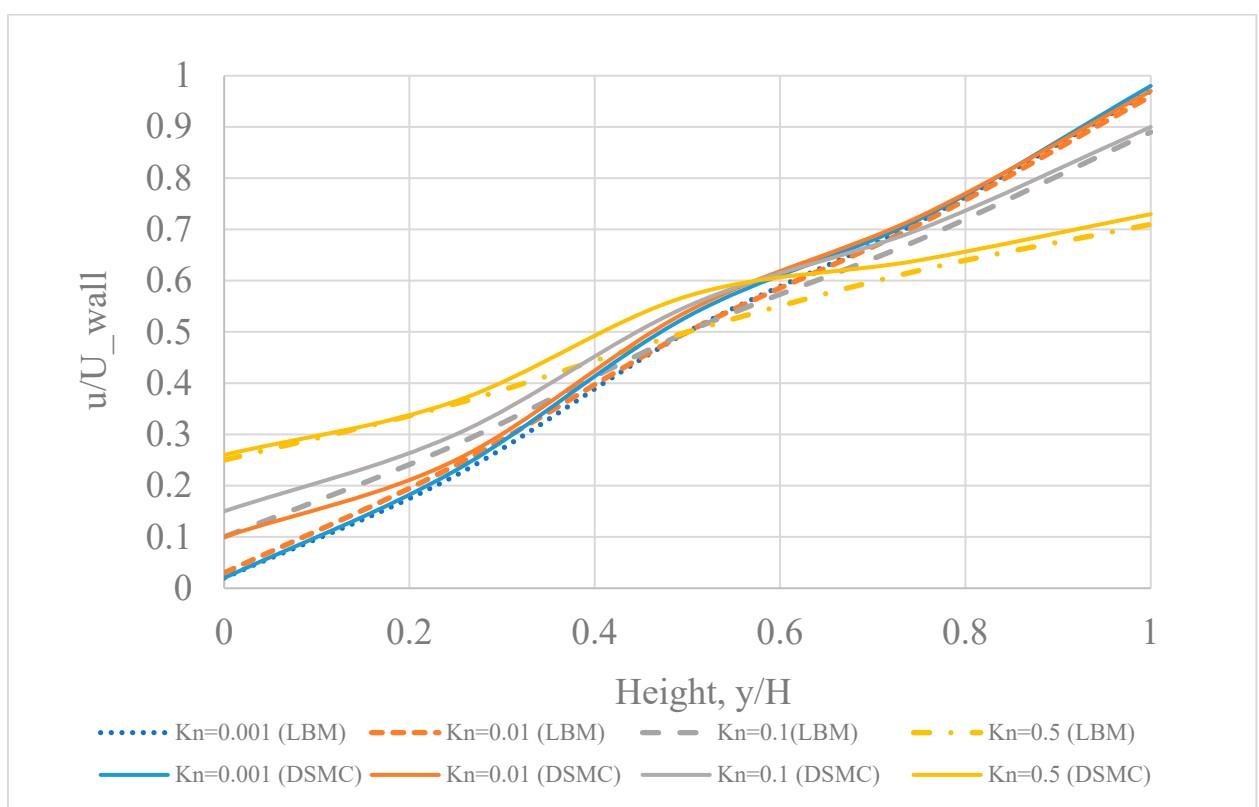

**Figure 6.** Validation of DSCM code by comparison with LBM results [25,32,33].

Figure 5 reveals the magnitude of slip velocity with the increase in the Knudsen number. It shows that the slip velocity magnitude was quite considerable and approached ~ 30% of upper-wall speed. In this flow regime, the molecular mean free path was either comparable to or larger than the distance between plates, which reduced the probability of intermolecular collisions and increased the probability of molecule-to-wall collisions. Due to higher slip velocity at both walls, the slip velocity profile flattened for *Kn* > 0.5 compared to lower Knudsen number cases.

Figure 6 shows that the DSMC results and results obtained by LBM for the same value of Knudsen number are within the acceptable error margin.

The convergence study for increasing the number of DSMC particles is presented in Appendix A.

### 3. Model of Deposition of Carbon Particles at Fibers

In this section, structured sets of fibers with a round cross-section (see Figure 1) in the CVD reactor are considered for non-zero Knudsen numbers. The present model does not account for the surface diffusion or re-emission of deposited species that can be accounted for in future research.

*3.1. The Reactor Flow Field*

The reactor flow field (Figure 7) was obtained by the FVM model for a CVD reactor with a bundle of fibers located at the center of reactor [4]. In Figure 7, the continuous CFD model is described in detail. Pressure, temperature, and velocity were calculated by the FVM at the middle of reactor, 4 mm upstream of fiber bundles, and used as the inlet boundary condition for the DSMC domain. By using the FVM, the concentration of species in feedstock gas flow in reactor was obtained in a prior study of the authors [4]. The DSMC domain encompasses the substrate (fibers), where the major feedstock gas species, which is responsible for surface CVD reaction, is $C_2H_2$.

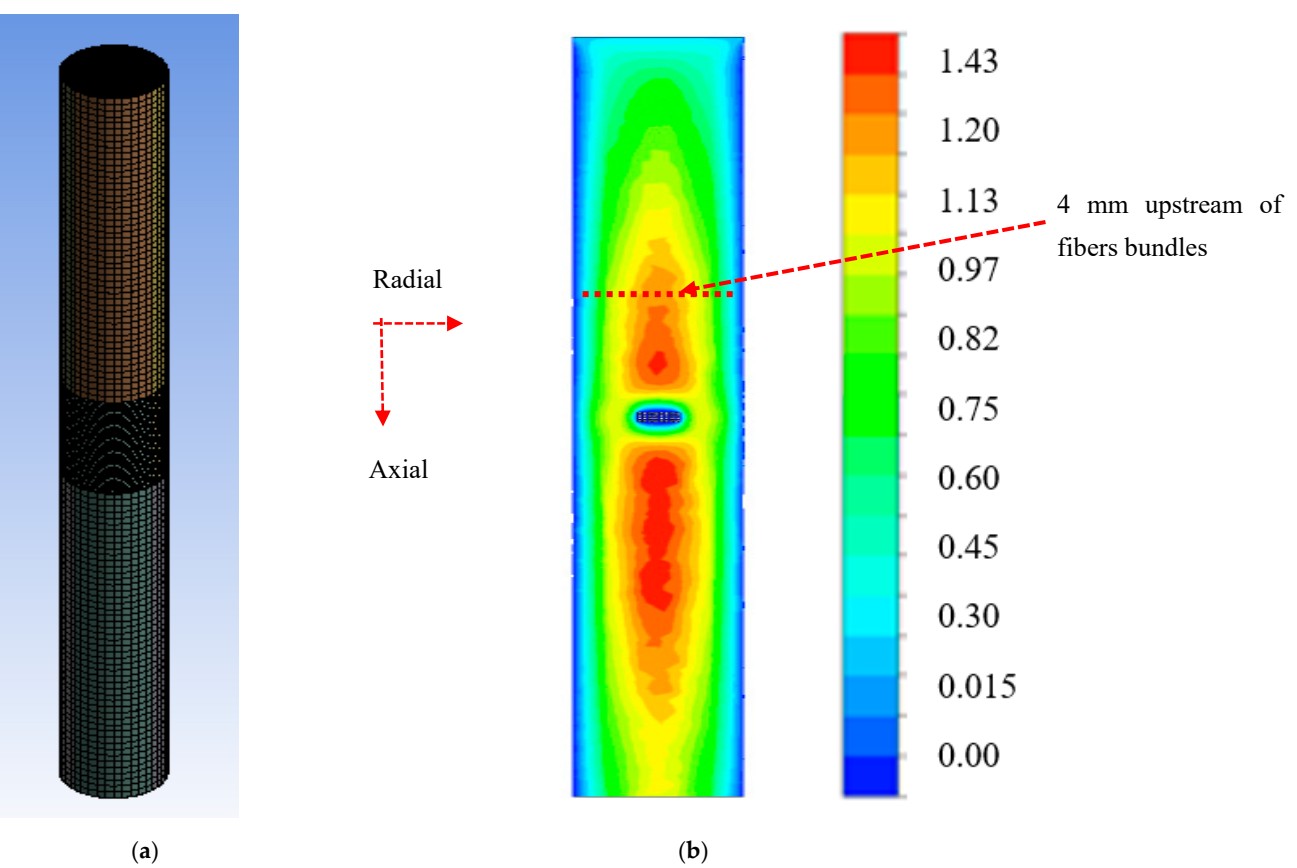

**Figure 7.** Chemical vapor deposition reactor [4]: (**a**) FVM mesh for CVD reactor and (**b**) isolines of velocity magnitude, m/s, along the vertical plane of the CVD reactor. The inlet boundary of the DSMC domain is shown.

The inlet velocity of the reactor was 1.28 m/s, with an inlet feedstock gas temperature of 25 °C. The static pressure inside the reactor was 1600 Pa and the corresponding Knudsen number of 0.002 for mean free path is $\lambda = 1.87 \times 10^{-5}$m, which represents the continuum

flow field. The goal of FVM simulations is to capture the feedstock gas flow velocity upstream of the bundle of fibers. The velocity was captured 4 mm upstream of the fiber bundles at the reactor centerline to be used for DSMC modeling (see Figure 7). To justify the location of the upstream boundary, a parametric study was conducted in which velocity was captured at distances of 1, 2, 3, 4, 5, 6, 7, 8, 9 and 10 mm upstream of the fiber bundles. With the range of 1 mm to 3 mm upstream of the fiber bundles, velocity is lower than at locations above 3 mm due to the formation of a stagnation zone over the fibers' surfaces. Beyond 4 mm upstream of the fibers' bundle, the feedstock gas velocity is no longer affected by the presence of fibers. Thus, the cross-section at 4 mm upstream of the fibers' bundle was chosen to export the CVD gas velocity from the FVM domain to the DSMC domain.

### 3.2. Setup of Fibers

In Figure 1, the fibers' set up is shown. The diameter, $d$, of fibers was 1 μm unless specified otherwise in subsequent sections, and the molecular mean free path of the gas was $2.5 \times 10^{-7}$ m, which corresponds to $Kn = 0.25$. Recall that the range $Kn > 0.1$ corresponded to the rarefied zone computed by DSMC. The DSMC domain length was 10 μm ($10d$) while the domain height was 6 μm ($6d$).

The size of the fibers' domain was much smaller compared to the length of the reactor. The end effect of the fibers can be neglected because the fiber length was much larger than the fibers' diameter. The side surfaces of the DSMC domain were taken as symmetrical boundaries to account for the fact that the bundle structure was self-repeating.

In DSMC modeling (see Section 2.1), the first step is to populate the domain with simulated particles, where each simulated particle represents billions of real molecules. The inlet velocity of the DSMC domain was imported from the FVM computations and was equal to 0.97 m/s for current computations (see Figure 7). This velocity was added to the molecular browning motion velocity of DSMC particles. The number density of gas was $1 \times 10^{25}$, and the number of DSMC cells was 1800. Each simulated particle represented $10^{12}$ real gas molecules.

At the fibers' surface, a combined diffuse and reflective boundary condition was considered, where the probability of reflection is $p$ and the probability of sticking to the fiber surface is $1-p$ (see Section 2).

Figure 8 shows that in the interior of the array of fibers in a bundle, the fluid velocity magnitude was smaller, ~0.33 m/s, compared to the inlet velocity of ~0.97 m/s. Between two fibers, the velocity increased due to the partial blockage of the flow area following the conservation of mass.

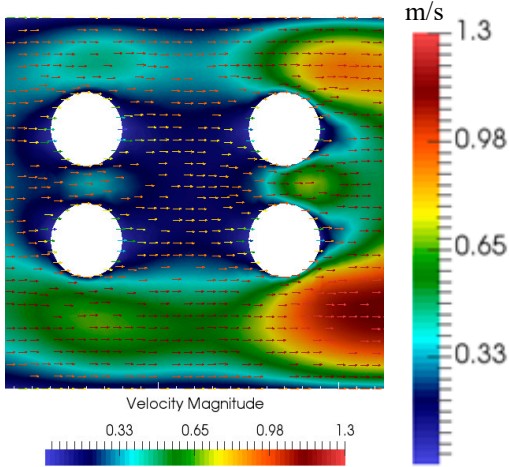

**Figure 8.** Velocity field around fibers in bundle computed by DSMC.

Figure 9 shows the intensity with which simulated particles are deposited at the fibers' surface. Particles, which flow from the inlet of the DSMC domain, start to deposit at an

individual fiber surface after particles reach it. Observation of Figure 8; Figure 9 shows that the first column of the fiber reached a steady state in terms of deposition rate more rapidly compared to the second-column fiber. While after 4 µs the number of deposited particles did not change with respect of time (Figure 9), for the second-column fiber it took ~8 µs (Figure 9) to reach the nearly steady-state phase of deposition. The steady-state number of deposited particles was different by almost 50% between the first and the second-column fibers (Table 2). This is explained by the fact that the second-column fibers were exposed to low-speed flow between fibers while the first-column fibers were exposed to the larger inlet velocity.

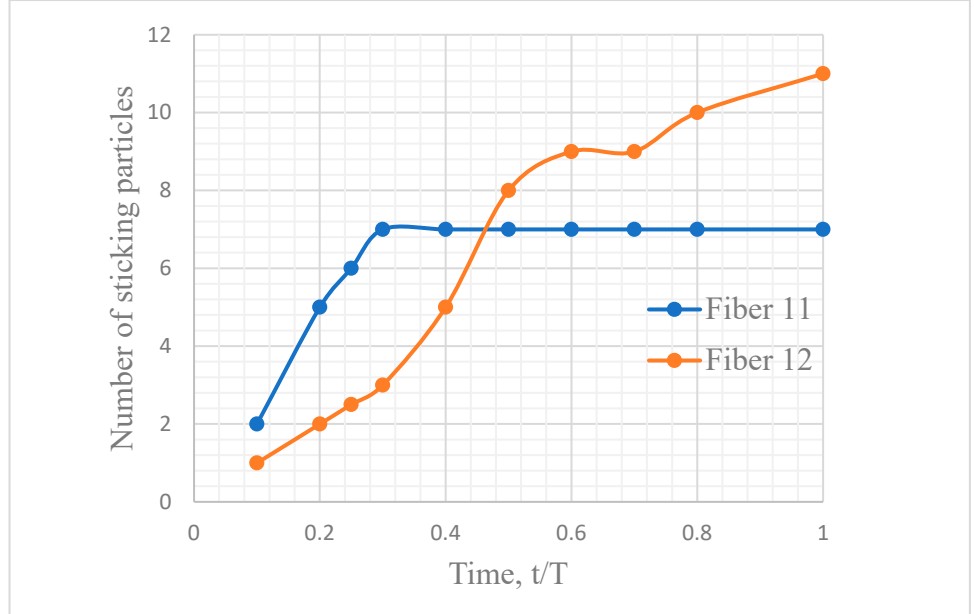

**Figure 9.** Particle deposition intensity over time at fiber surface—Fiber 11 and Fiber 12—for 50% sticking coefficient. The normalized time, $t/T$, is used, where $T = 10$ µs.

**Table 2.** Number of deposited particles over fiber surfaces.

| Fiber | Number of Simulated Particles | Mass of Deposited Layer, Nanogram |
|---|---|---|
| Fiber 11 | ~7 | 290 |
| Fiber 12 | ~11 | 457 |
| Fiber 21 | ~7 | 290 |
| Fiber 22 | ~11 | 457 |

Table 2 shows how carbon deposited over the fiber surfaces for the regular orientation of fibers (Figure 1), with $Kn = 0.25$. The molar mass of $C_2H_2$ was 26.04 g/mol, after subtracting two atoms of $H_2$; as hydrogen does not participate in surface deposition, the molecular mass of remaining $C_2H_2$ was $4.16 \times 10^{-23}$ kg. As each DSMC particle consisted of $10^{12}$ molecules, the mass of each DSMC particle would be $4.16 \times 10^{-11}$ kg.

A comparison to experiments for low Knudsen numbers is available in a prior study of the authors ([4]; see Figure 4), in which the carbon deposition rate obtained by CFD simulations was compared to the experiments [35]. The experiments describe deposition on a bundle of fibers, with the diameter of an individual fiber being 10 µm. The reactor and feedstock gas conditions were the same as in the present study. The fibers' sample was tied together and folded four times. The sample therefore had the form of an "eight"; see Ref. [35], page 30, Figure 3.2. The present study helps to design future experiments using the same reactor and CVD process in which individual fibers of micron-scale diameters are separated and form a structured matrix.

### 3.3. Effect of Sticking Coefficient on Deposition Rate

In the previous section, simulations were conducted with the sticking coefficient equal to 50%. To understand the role of the value of the sticking coefficient, a parametric study is conducted in the current section for a range of sticking coefficients, $\eta$, namely, 20%, 40%, 50%, 60% and 80%. With a higher value of the sticking coefficient, particles would have a higher probability of sticking to the fiber surface, while a lower probability of sticking repels more particles away from the fiber surface. The simulations conducted in the current section quantify the trend.

Figures 10 and 11 show how the particle deposition rate varies with respect to the value of the sticking coefficient and with respect to the position of the fiber in the bundle. Front-column fibers, which face the flow stream first, have a lower deposition rate compared to the deposition at the surface of second-column fibers. A higher sticking coefficient provides a higher deposition rate while a lower sticking coefficient provides a lower deposition rate. This dependence is monotonic and non-linear, as detailed in Figure 12.

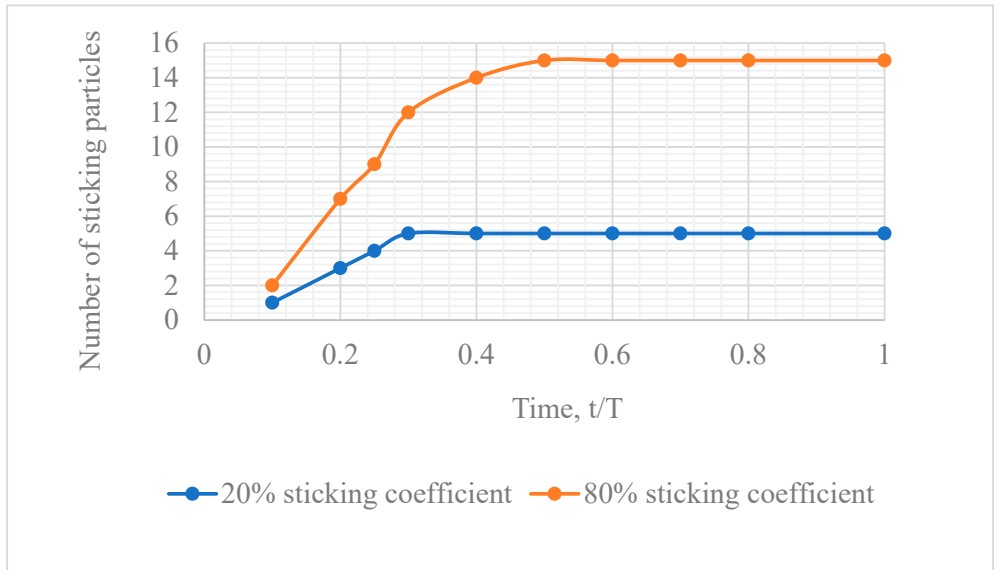

**Figure 10.** Carbon particle deposition on Fiber 11 for 20% sticking coefficient and 80% sticking coefficient (*Kn* = 0.25).

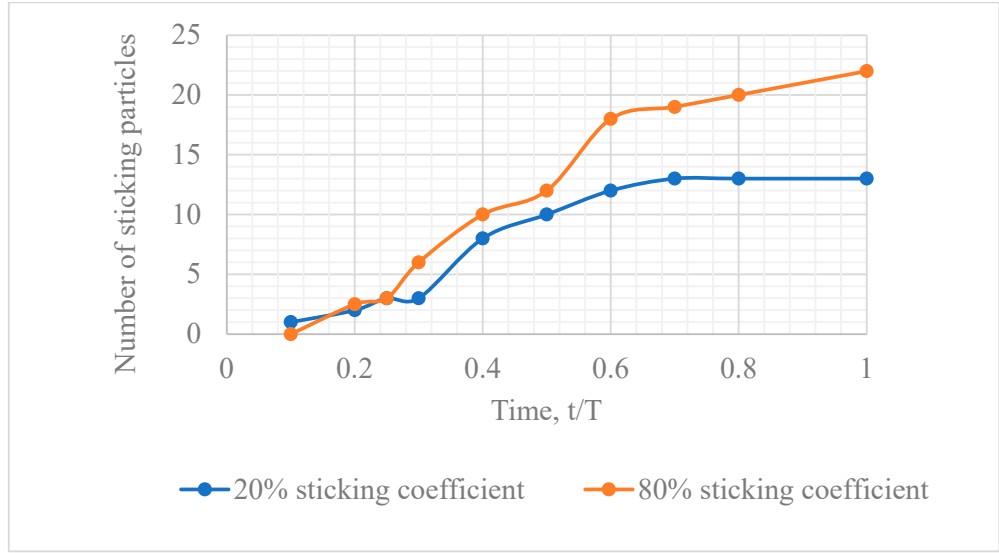

**Figure 11.** Carbon particle deposition on Fiber 12 (first row, second column fiber) for 20% sticking coefficient and 80% sticking coefficient (*Kn* = 0.25).

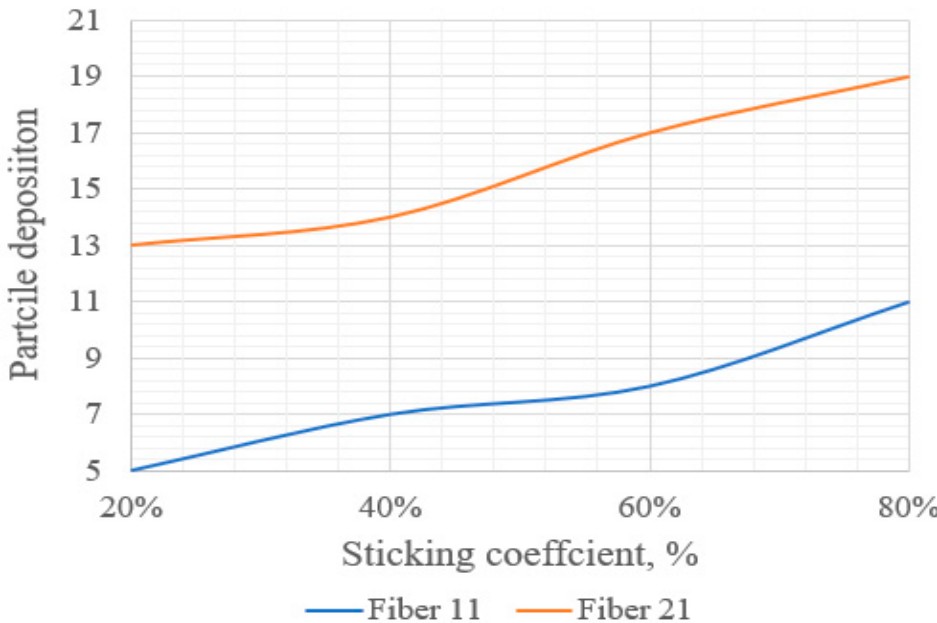

**Figure 12.** Particle deposition as a function of sticking coefficient at fiber surface.

In the present study, the surface reaction rate was estimated based on the species flux to the surface times the so-called sticking probability (or sticking coefficient). With this simplification, the only necessary input to the DSMC simulations was the sticking probability for each DSMC particle.

To understand the role of the value of sticking coefficient, a parametric study was conducted for a wide range of values of sticking coefficient, from 20% to 80%.

A few authors [22,31] have suggested incorporating reaction rate and activation energy to obtain the sticking coefficient in a more precise, physically accurate way. This will be addressed in the future development of the model proposed in the current study. Recent references have stressed the importance of the use of sticking coefficient. As per Ref. [36], "we adopt a commonly simplified approach—the sticking coefficient (SC) method—to predict the deposition rate when the process is considered to be surface deposition limited". as per Ref. [37], "Central in the idea of the line-of-sight models is the (effective) sticking coefficient which is defined as the ratio of the number of molecules of species that become part of film with a rate, to the total number of impinging molecules, per area per unit of time. . . .. For Monte Carlo methods, the molecules reaching the substrate are adsorbed according to the sticking coefficient S and reflected diffusively with a probability $(1 - S)$".

*3.4. Effect of Distance between the Fibers*

The distance between the centers of fibers plays a vital role in the deposition magnitude and profile. It will be shown in this section that for the same volume fraction of fibers in a bundle, the closer the distance between fibers, the larger the particles' deposition.

Figures 13 and 14 show that, for the same volume fraction of fibers, changing the distance between the fibers affects the deposition pattern and magnitude. A numerical study was conducted in which the volume fraction of fibers was kept constant while changing the diameter and distance between the fibers. Two cases were considered in the study. For case 1, the diameter of fibers was 0.25 μm and the distance (between the circumference) between the fibers was 0.5 μm. For case 2, the diameter of fibers was 0.5 μm and the distance between the fibers was 1 μm. For both cases, the volume fraction was 20%. The flow Knudsen number was 0.25. As the diameters of fibers were chosen according to volume fraction of fibers, the mean free path of gas was changed to keep the Knudsen number constant. For case 1, the mean free path of $C_2H_2$ was $6.25 \times 10^{-8}$ m, while for case 2, it was $1.25 \times 10^{-7}$ m.

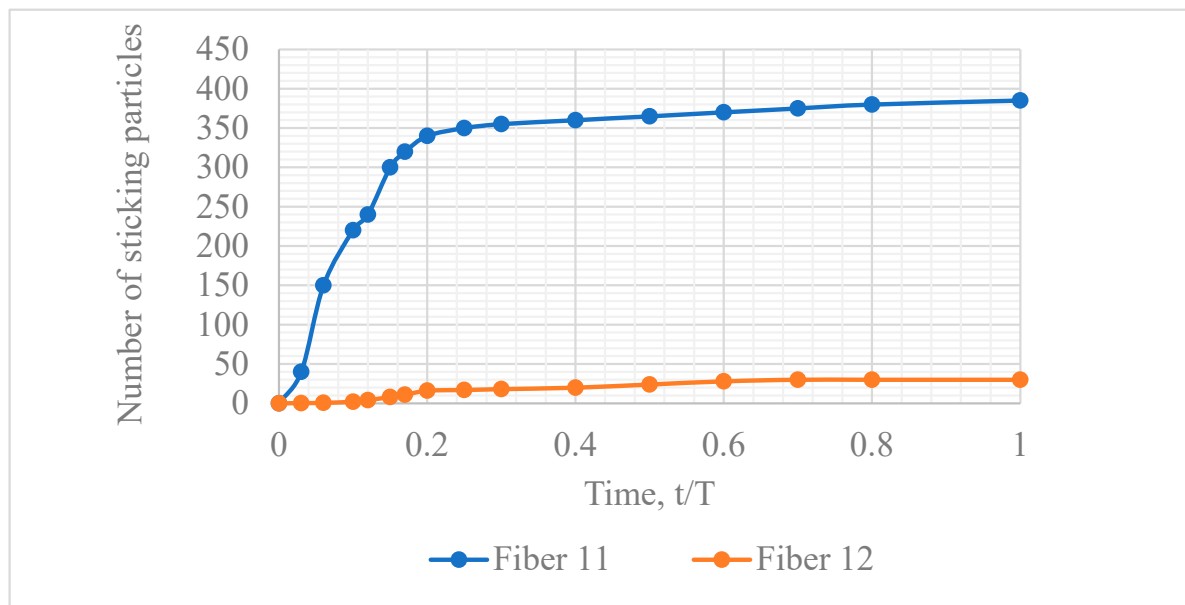

**Figure 13.** Particle deposition rate for 0.25 μm diameter fiber at 0.5 μm distance between fibers (case 1) for Fiber 11 and Fiber 12 (*Kn* = 0.25). The volume fraction of fibers was 20% and sticking coefficient was 20%.

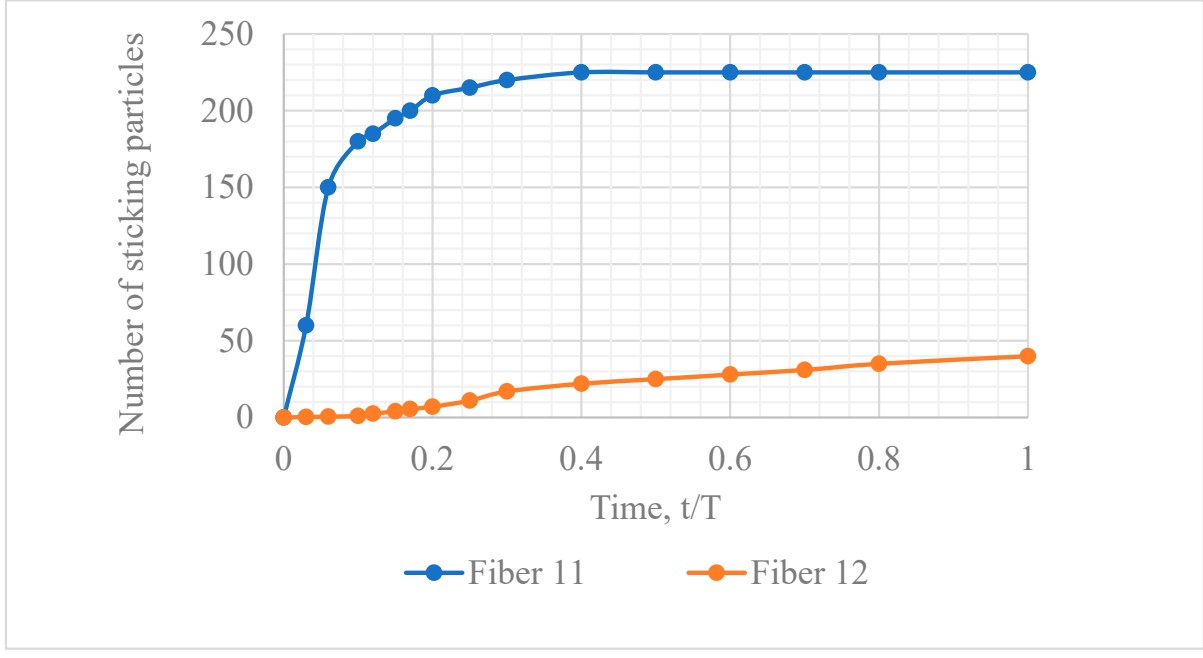

**Figure 14.** Particle deposition rate for 0.5 μm diameter fiber at 1 μm distance between fibers (case 2). The volume fraction and sticking coefficient were the same as in Figure 13.

For case 1, by time moment $t/T = 0.25 \sim 360$, simulated particles were deposited at the fibers' surface (see Figure 13), while for case 2 with the same 20% volume fraction, at $t/T = 0.40$, 200, simulated particles were deposited. It shows that the smaller distance between the fibers and smaller diameter of fibers causes ~1.63 times larger deposition for front-column fibers. For the second-column fibers, the magnitude of deposition becomes reversed. Due to the higher distance, fibers of 0.5 μm diameter and 1 μm distance have 1.5 times larger deposition than a 0.25 μm diameter and 0.5 μm distance. Figures 13 and 14

show that, compared with front fibers, a rear fiber accumulates particles at a slower rate over a longer time interval.

### 3.5. Effect of Rarefaction

Rarefaction in gas affects the deposition profile at fibers' surface. Rarefaction happens either due to lower pressure or due to a smaller diameter of fibers. With a higher Knudsen number (Equation (1)) caused by a smaller fiber diameter with the same volume fraction of fibers in a bundle, gas particles must travel longer to collide due to a larger molecular mean free path. Also, the slip velocity increases with the Knudsen number. With a higher slip velocity, flow resistance is lower due to a smaller velocity gradient normal to a rigid surface and, consequently, reduced skin friction. To quantify the residence time of particles in the vicinity of fibers, an area of influence (Figure 15a) was considered around the bundle of fibers. The length and height of the area of influence were 6d, in microns (square-shaped), respectively. The diameter, d of fibers was 0.25 microns and the distance between the fibers was 0.50 microns (case 1).

For the no-slip condition (Figure 15b), at the surface of fibers all particles which did not stick to the fibers' surface left the domain within 2 microseconds ($0.2 \times 10$ μs). For $Kn = 0.3$ (Figure 15c), where the gas flow partially slipped at the fibers' surface, the residence time was smaller due to lower friction resistance at the fibers' surface. Particles, which did not stick to the fiber surface left the domain within $0.07 \times 10$ μs $= 0.7$ μs. For $Kn = 0.5$ (Figure 15d), the residence time decreased to $0.06 \times 10$ μs $= 0.6$ μs. With a lower residence time, the available time for particles to participate in surface deposition decreases and the amount of deposition is lower compared to no-slip conditions.

Figure 16 shows that with the increase in the Knudsen number, the residence time decreased nearly linearly for $Kn > 0.1$. The residence time was defined as such that 90% of particles had left the domain. For smaller Knudsen numbers including the continuum zone ($Kn = 0.001$), the residence time was close to 2 μs (see also Figure 15b). In terms of physical significance, with a smaller fiber size, the residence time would be smaller due to the slip velocity at wall, which would offer less time for surface reactions. With less time available for surface reaction, and the smaller surface area available for deposition, the CVD process, which is a relatively slow process itself, would produce less solid deposition in the substrate surface.

In Figure 17, 0 degrees and 180 degrees represent equatorial points, and 90 degree represents the stagnation point. The asymmetric deposition with respect to the centerline of each fiber is caused by the asymmetry of the set-up, where each fiber has neighboring fiber on the one side and domain boundary on the other side (see Figure 1).

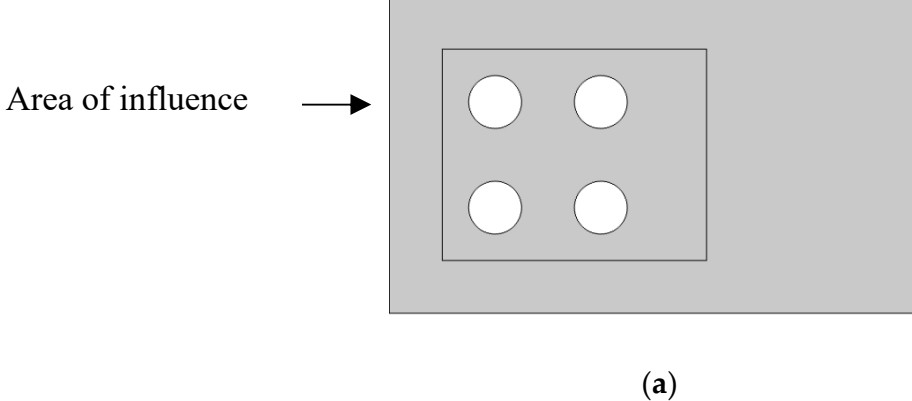

**(a)**

**Figure 15.** *Cont.*

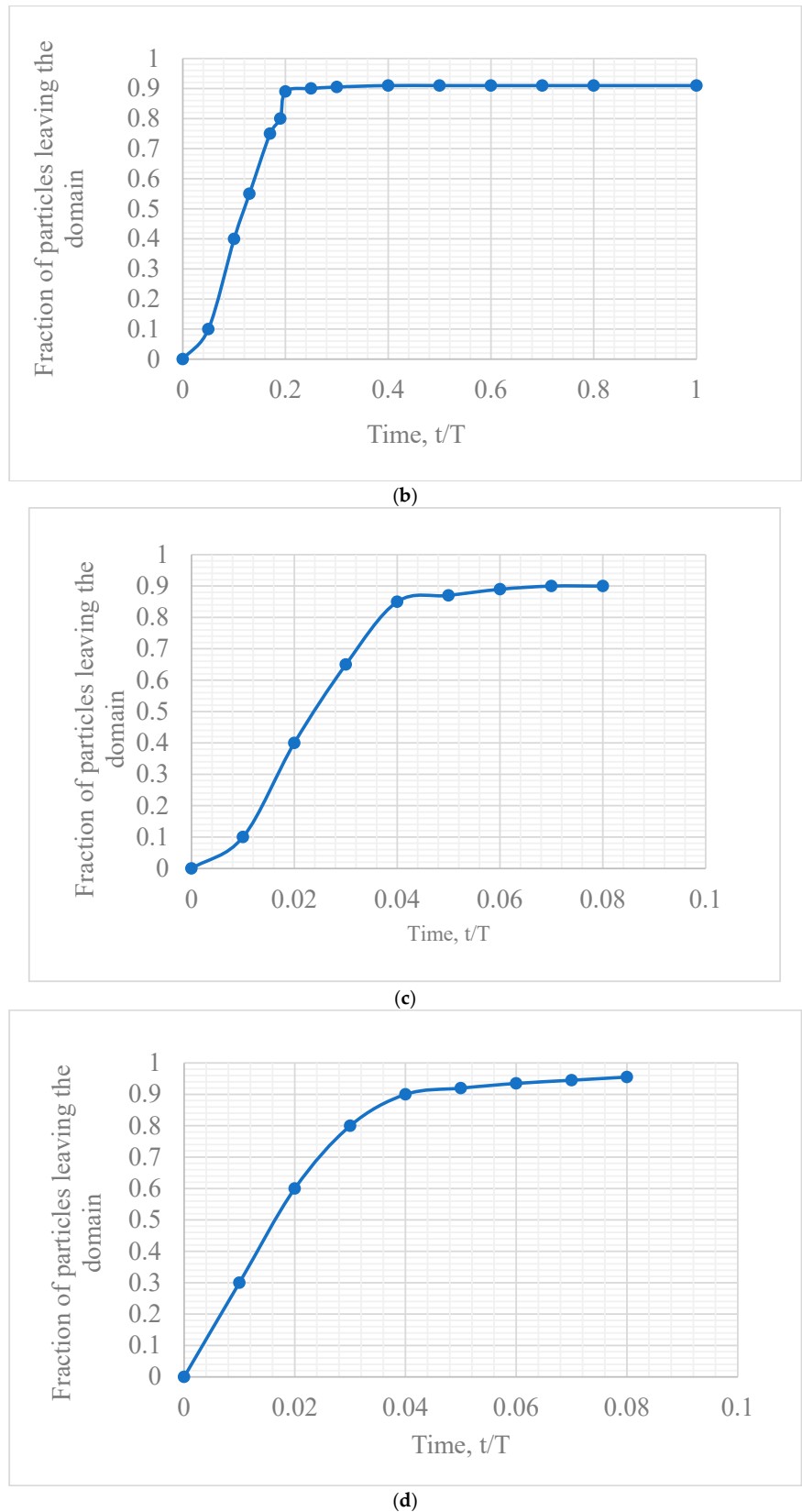

**Figure 15.** The fraction of particles leaving the domain in flow across the set of fibers for sticking coefficient equal to 50% for the range of Knudsen numbers: (**a**) area of influence around the bundle of fibers; (**b**) no-slip condition; (**c**) $Kn = 0.3$ and (**d**) $Kn = 0.5$. The considered time interval $T = 10$ μs.

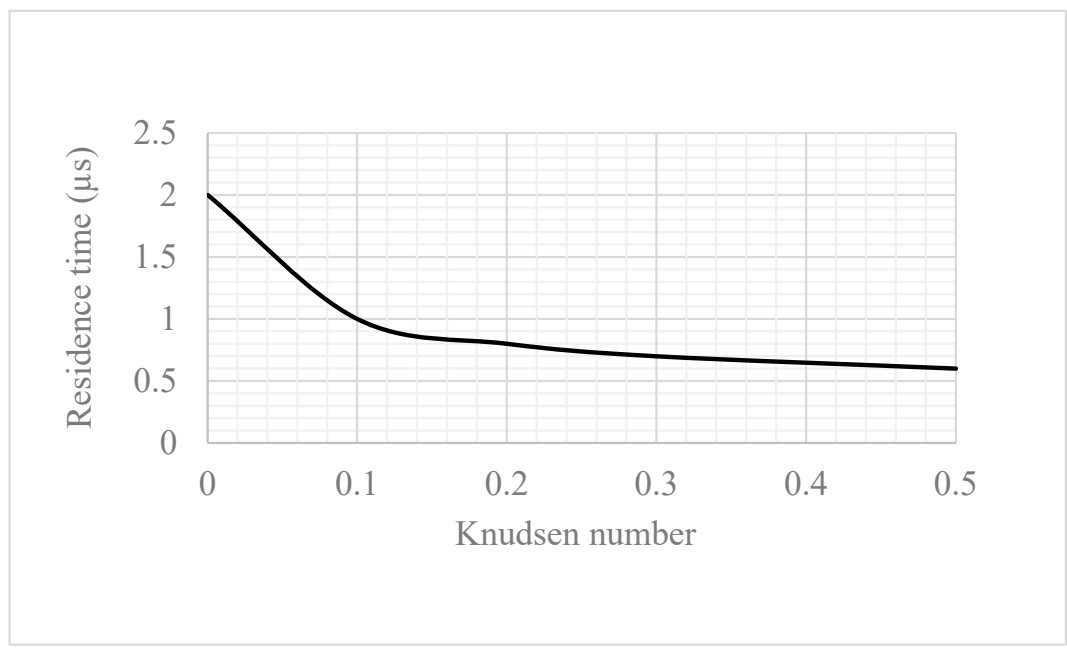

**Figure 16.** Residence time vs. Knudsen number.

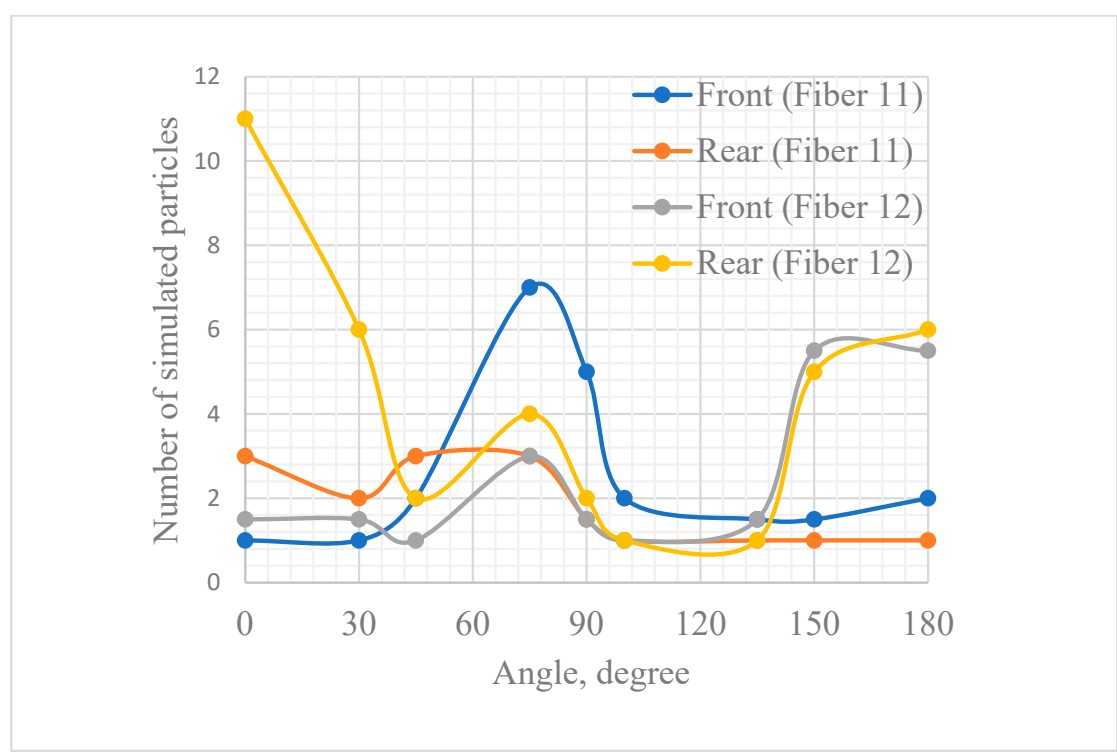

**Figure 17.** Particle deposition profile at Fiber 11 and Fiber 12 for Kn = 0.3.

Figure 17 shows the deposition profile along the fiber circumference modeled by DSMC method. In the front side of fiber with respect to flow (for angles between 45° and 135°), deposition is larger compared to the rear side for Fiber 11. For Fiber 12, both front and rear semi-circumferences have larger deposition of particles compared to Fiber 11. Due to the low-speed wake downstream of Fiber 11 (see Figure 8), CVD gas has a larger residence time near the second fiber (Fiber 12) that allows larger and more uniform deposition compared to the first-column fibers.

## 4. Conclusions

The goal of this study was to adopt Direct Simulation Monte Carlo (DSMC) to evaluate the CVD rate for fibers of nano-scale diameter by using simulated $C_2H_2$ particles. For micro- and nano- scale fibers with a non-zero Knudsen number, the conventional finite volume method (FVM) is not applicable, while molecular methods like DSMC are very computationally intensive to model flowfield for an entire industrial reactor. This study couples FVM with DSMC, where the bulk flow in a reactor and its chemical composition were computed by FVM and flow near the fibers is modeled by DSMC.

With a reduction in fiber diameter, the corresponding Knudsen number increases and velocity at the fibers' surface starts to deviate from no-slip to slip velocity mode. Due to slip velocity at the fibers' surface, the residence time of particles representing reactive molecules within the fibrous medium decreases that eventually decreases the amount of surface deposition at the fibers' surface. The deposition magnitude appears to depend strongly upon the sticking coefficient. For the 80% sticking coefficient, the deposition rate is nearly doubled compared to that for the 20% sticking coefficient. Fibers at the front show a higher deposition rate compared to fibers at the second column due to longer contact with feedstock gas compared to the second-column fibers.

The proposed methodology will be extended to unstructured set-ups of fibers with multiple diameters in future research.

**Author Contributions:** Conceptualization, A.P. and H.B.; methodology, H.B. and A.P.; software, H.B.; validation, H.B.; investigation, A.P.; data curation, H.B.; writing—original draft preparation, H.B.; writing—review and editing, A.P.; visualization, H.B.; supervision, A.P.; project administration, A.P. All authors have read and agreed to the published version of the manuscript.

**Funding:** This research received no external funding.

**Data Availability Statement:** The data presented in this study are available on request from the corresponding author. The data are not publicly available due to privacy.

**Conflicts of Interest:** The authors declare no conflict of interest.

## Appendix A  Convergence Study

A total of 40,000 simulated DSMC particles were chosen for the whole domain that corresponds to 100 particles per cell. This is five times larger than the previous simulation case depicted in Figure 4. With the increased number of particles, the velocity profile has a much smaller amplitude of oscillations, as depicted in Figure A1.

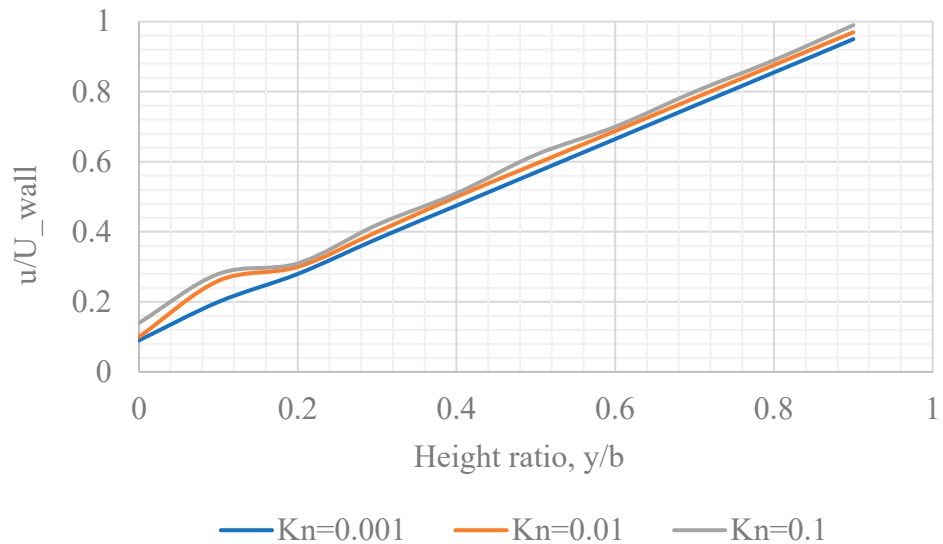

**Figure A1.** Normalized velocity profile for Couette flow.

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
