# Peer review of "Continuum and Molecular Modeling of Chemical Vapor Deposition at Nano-Scale Fibrous Substrates"

_mca, doi:10.3390/mca28060112_

Round 1
Reviewer 1 Report
Comments and Suggestions for Authors
Overall, the paper is well-written and the methodology is sound. However, there are several areas that could be improved before the paper can be published:
1. The paper relates the reduced deposition rate to the change of the residence time due to higher slip velocity as a result of higher Kn number at smaller fiber diameter. Does the conclusion consider the smaller surface area available for deposition as one of the possible reasons that lead to the drop in redeposition rate?
2. While the authors describe the DSMC model and simulation setup in great details, more information are needed on their CFD model set-up, boundary conditions, mesh, etc. Currently the CFD model is described very briefly compared to the extensive DSMC details.
3. The discussion of the computational Methodology of DSMC in section 2. is rather lengthy and contains the convergence study which is useful for the reader but more suitable to be put into the appendix.
4. Comparison to experimental data, if available, would significantly strengthen the paper. There are no discussions on experimental data that points to the finding of the simulation, especially on the influence of smaller fiber diameter on the deposition rate. While specific experiments might not be available, the authors should at least consider some discussion along this direction and how to verify their findings from experiments.
5. The model does not account for surface diffusion or re-emission of deposited species, which may be relevant. Can the authors elaborate on these processes?
6. Label of fig. 1 is incomplete, only two fibers 11 and 12 are shown, while it was described in the paper how these fibers are indexed, it would be useful to have a complete schematic for the reader.
7. The caption of Fig. 16 is not consistent with the plot, as the figure is showing the fraction of particles leaving the domain, not the residence time. Also, residence time appears to be defined as 90% of particles leaving the domain based on the discussion in the paper, but it is not clearly explained in the paper. Fig. 16 (c-d) shows x axis for t/T between 0 and 0.1 for T=10us, which does not translate the residence time of 6us or 7us as shown in the discussion, but 0.6 and 0.7 us instead. A typo “Hear” should be corrected.
8. Fig. 17 does not have the unit of the residence time. Is it in milliseconds?
9. The discussion on the Fig.18 appears to be missing one figure as the discussion includes (a) and (b) while only one figure is shown in the current paper.
1 Figure 18 shows the spatial deposition profile around the fiber circumference due to flow orientation. Some discussion on the cause of the asymmetric deposition would be beneficial.
1 The velocity field of Fig.9 do not point to any obvious flow structures like vortices or recirculation that could preferentially increase deposition on Fiber 12. The authors need to explain why the Fiber 12 has a much higher deposition. Also, the flow field in Fig. 9 does not appear perfectly symmetric, are they coming from possible statistical noise.
Comments on the Quality of English Language
The paper titled "Continuum and Molecular Modeling of Chemical Vapor Deposition at Nano-scale Fibrous Substrates" presents a novel coupling of computational fluid dynamics (CFD) and direct simulation Monte Carlo (DSMC) methods to model chemical vapor deposition (CVD) around nano-scale fibrous substrates. The authors use CFD to model the bulk reactor flow and extract inlet conditions for a more detailed DSMC model around the fibers. They validate their DSMC code for Couette flow and explore the effects of fiber spacing, sticking coefficient, and rarefaction on deposition rate and particle residence time. Their multiscale modeling approach shows promise for enabling high-fidelity simulations for emerging nanoscale applications.
Reviewer 2 Report
Comments and Suggestions for Authors

Round 2
Reviewer 1 Report
Comments and Suggestions for Authors
Thanks for forwarding the revised manuscript and the cover letter from the authors. After reviewing the new version, I believe the authors have addressed all the questions from my earlier review and therefore, I would recommend the publication of the current paper.
Reviewer 2 Report
Comments and Suggestions for Authors
Dear authors,
thank you for listening to my wishes. I am sure that the resulting publication will be of interest to a wide range of students and researchers.
Sincerely,
reviewer.